# Immunological Aspects of SARS-CoV-2 Infection and the Putative Beneficial Role of Vitamin-D

**DOI:** 10.3390/ijms22105251

**Published:** 2021-05-16

**Authors:** Ming-Yieh Peng, Wen-Chih Liu, Jing-Quan Zheng, Chien-Lin Lu, Yi-Chou Hou, Cai-Mei Zheng, Jenn-Yeu Song, Kuo-Cheng Lu, You-Chen Chao

**Affiliations:** 1Division of Infectious Disease, Department of Medicine, Taipei Tzu Chi Hospital, Buddhist Tzu Chi Medical Foundation, New Taipei City 231, Taiwan; 16044@s.tmu.edu.tw; 2Division of Nephrology, Department of Medicine, Taipei Hospital, Ministry of Health and Welfare, New Taipei City 242, Taiwan; wayneliu55@gmail.com; 3Graduate Institute of Clinical Medicine, College of Medicine, Taipei Medical University, Taipei 11031, Taiwan; jingquan235@gmail.com (J.-Q.Z.); Athletics910@gmail.com (Y.-C.H.); 4Division of Pulmonary Medicine, Department of Internal Medicine, School of Medicine, College of Medicine, Taipei Medical University, Taipei 11031, Taiwan; 5Division of Pulmonary Medicine, Department of Internal Medicine, Shuang Ho Hospital, Taipei Medical University, New Taipei City 23561, Taiwan; 6Division of Nephrology, Department of Medicine, Fu Jen Catholic University Hospital, School of Medicine, Fu Jen Catholic University, New Taipei City 242, Taiwan; janlin0123@gmail.com; 7Division of Nephrology, Department of Medicine, Cardinal-Tien Hospital, School of Medicine, Fu-Jen Catholic University, New Taipei City 234, Taiwan; 8Taipei Medical University-Research Center of Urology and Kidney (TMU-RCUK), Taipei Medical University, Taipei 110, Taiwan; 11044@s.tmu.edu.tw; 9Division of Nephrology, Department of Internal Medicine, Taipei Medical University Shuang Ho Hospital, New Taipei City 235, Taiwan; 10Division of Nephrology, Department of Internal Medicine, School of Medicine, College of Medicine, Taipei Medical University, Taipei 110, Taiwan; 11Division of Cardiovascular Surgery, Department of Surgery, Taipei Tzu Chi Hospital, Buddhist Tzu Chi Medical Foundation, New Taipei City 231, Taiwan; ttwyl123@yahoo.com; 12School of Medicine, Tzu Chi University, Hualien 970, Taiwan; thanthanwinnge@gmail.com; 13Division of Nephrology, Department of Medicine, Taipei Tzu Chi Hospital, Buddhist Tzu Chi Medical Foundation, New Taipei City 231, Taiwan; 14Division of Gastroenterology, Department of Internal Medicine, Taipei Tzu Chi Hospital, Buddhist Tzu Chi Medical Foundation, New Taipei City 231, Taiwan

**Keywords:** coronavirus disease 2019, severe acute respiratory syndrome coronavirus 2, vitamin D, adaptive immunity, innate immunity, angiotensin-converting enzyme 2, renin–angiotensin–aldosterone system

## Abstract

Coronavirus disease 2019 (COVID-19), caused by severe acute respiratory syndrome coronavirus-2 (SARS-CoV-2) is still an ongoing global health crisis. Immediately after the inhalation of SARS-CoV-2 viral particles, alveolar type II epithelial cells harbor and initiate local innate immunity. These particles can infect circulating macrophages, which then present the coronavirus antigens to T cells. Subsequently, the activation and differentiation of various types of T cells, as well as uncontrollable cytokine release (also known as cytokine storms), result in tissue destruction and amplification of the immune response. Vitamin D enhances the innate immunity required for combating COVID-19 by activating toll-like receptor 2. It also enhances antimicrobial peptide synthesis, such as through the promotion of the expression and secretion of cathelicidin and β-defensin; promotes autophagy through autophagosome formation; and increases the synthesis of lysosomal degradation enzymes within macrophages. Regarding adaptive immunity, vitamin D enhances CD4^+^ T cells, suppresses T helper 17 cells, and promotes the production of virus-specific antibodies by activating T cell-dependent B cells. Moreover, vitamin D attenuates the release of pro-inflammatory cytokines by CD4^+^ T cells through nuclear factor κB signaling, thereby inhibiting the development of a cytokine storm. SARS-CoV-2 enters cells after its spike proteins are bound to angiotensin-converting enzyme 2 (ACE2) receptors. Vitamin D increases the bioavailability and expression of ACE2, which may be responsible for trapping and inactivating the virus. Activation of the renin–angiotensin–aldosterone system (RAS) is responsible for tissue destruction, inflammation, and organ failure related to SARS-CoV-2. Vitamin D inhibits renin expression and serves as a negative RAS regulator. In conclusion, vitamin D defends the body against SARS-CoV-2 through a novel complex mechanism that operates through interactions between the activation of both innate and adaptive immunity, ACE2 expression, and inhibition of the RAS system. Multiple observation studies have shown that serum concentrations of 25 hydroxyvitamin D are inversely correlated with the incidence or severity of COVID-19. The evidence gathered thus far, generally meets Hill’s causality criteria in a biological system, although experimental verification is not sufficient. We speculated that adequate vitamin D supplementation may be essential for mitigating the progression and severity of COVID-19. Future studies are warranted to determine the dosage and effectiveness of vitamin D supplementation among different populations of individuals with COVID-19.

## 1. Introduction

Coronavirus disease 2019 (COVID-19), a rapidly spreading respiratory illness caused by the severe acute respiratory syndrome (SARS) coronavirus 2 (SARS-CoV-2) virus [1], constitutes a global health emergency. Notably, immune responses activated by the coronavirus, including adaptive immune responses in the earlier and asymptomatic stages, prevent further disease progression. Patients’ immune status plays a pivotal role in predicting their prognosis. Patients with immunodeficiency or aberrant immunity may promote viral replication and subsequent tissue damage with multiple organ failure. At the other end of the spectrum, overactive immune responses are correlated with immunopathological conditions and further tissue destruction.

Vitamin D insufficiency or deficiency is a highly prevalent global problem, affecting over a billion people worldwide [2,3]. In the COVID-19 era, it is interestingly found that vitamin D deficiency related with higher risks for SARS-CoV-2 infection [4,5]. It also associated with respiratory immune impairment and increased COVID-19 severity and mortality [6,7]. Moreover, vitamin D deficiency leads to cytokine storms [8] and to excessive tissue damage and mortality in patients with SARS-CoV-2 infection [6,9]. However, the clinical effects of vitamin D supplementation in COVID-19 era is still controversial [10,11,12]. It is critically important to understand how vitamin D influence the impact of COVID-19 and determine appropriate dosages of vitamin D among different patient population.

COVID-19 world map evidence the variations of COVID-19 occurrence, spread, severity, and mortality varied around the world, and this has been found to overlap with vitamin D deficiency areas. Understanding the immunological differences in individuals and populations is essential. With the continuing spread of infection, the lack of targeted therapy presents a major problem [2]. Adequate supplementation of vitamin D might be helpful for preventing those at higher risk of vitamin D deficiency (e.g., older adults, patients, individuals with comorbid conditions, and immunocompromised individuals) from contracting COVID-19. In this review, we will discuss the details of human immunity defense against SARS-CoV-2, molecular mechanisms underlying vitamin D-related anti-SARS-CoV-2 immunity and different clinical situations.

## 2. COVID-19 Pathophysiology

During the initial 1–2 days of infection, SARS-CoV-2 either passes through the upper respiratory tract or directly enters the lower respiratory tract, infecting both bronchial and alveolar cells [13,14]. Angiotensin-converting enzyme (ACE) 2 (ACE2), which is expressed in most human cells, acts as a receptor for SARS-CoV-2 [15]. In vitro studies have indicated that SARS coronavirus (SARS-CoV) primarily infects the ciliated cells which is high in ACE2 expression in the conducting airways [16], and that ACE2 expression is lower within other cells in these airways [13,17]. Local propagation of the virus occurs when innate immunity cannot be fully initiated. At this stage, the virus can be detected in nasal swabs. As the infection progresses, the virus moves through the respiratory tract, and a robust innate immune response is triggered. By the time the infection is clinically manifested, viremia has already occurred; that is, the virus has already entered the peripheral bloodstream [18,19]. As the disease progresses, the virus continues to affect other organs that mainly express ACE2 (e.g., the heart and its blood vessels, the kidneys, and the gastrointestinal tract). Thus, further disease progression and systemic organ damage tend to occur in patients with severe pulmonary symptoms [19]. Cytokine storms, characterized by strong inflammatory responses in response to immunological threats, are possibly responsible for systemic organ dysfunction. The clinical symptomatology of COVID-19 varies, from a lack of symptoms to the presentation of symptoms related to local infection (pneumonia), followed by either recovery or disease progression with systemic manifestations [20].

In almost 20% of patients infected with SARS-CoV2, the disease progresses to the point where pulmonary infiltrates are developed [1]—a point at which the inhaled virus reaches the terminal airway and mainly infects alveolar type II cells [21]. Most alveoli in the peripheral and subpleural regions that are infected [22] undergo substantial apoptosis as SARS-CoV2 propagates locally [23]. Pathological data from a case report indicated diffuse alveolar damage with subsequent infiltration of multinucleated giant cells and the formation of fibrin-rich hyaline membranes [19]. Notably, aberrant healing from SARS-CoV-2 results in more severe scarring and fibrosis compared with that from other types of acute respiratory distress syndrome (ARDS).

### 2.1. Viral Cell Entry

In order to enter the host cell, the SARS-CoV-2 firstly binds to the cell surface receptor for virus attachment, then enters the endosome, and finally fuses the viral membrane and the lysosomal membrane. The virus has several structural proteins, such as S (Spike), M (Membrane), N (Nucleocapsid), and E (Envelope) and HE (Hemagglutinin esterase). Protein S consists of 2 sub-units: S1 and S2. S1 plays a role in viral attachment. SS (signal sequence), NTD (terminal domain N), RBD (receiver link domain), and RBM (receiver link pattern) are parts of S1 proteins. The S2 subunit consists of the fusion peptide (FP), protease cleavage site (S2′), central helix (CH), connector domain (CD), heptad repeat (HR) 1 and 2, transmembrane domain (TM), and cytoplasmic tail (CT). The S2 subunit fuses the virus with the host cells [24].

On mature viruses, the spike protein exists in the form of trimers, and there are three S1 heads that bind to the receptor on the three-trimeric membrane fusion S2 stalk. SRAS-CoV S1 contains a receptor binding domain (RBD) which specifically recognizes the ACE2 as its receptor. RBD continuously switches between the standing position (for the connection of the receiver) and the lying position (to escape immunity) [25]. The furin preactivation of the spike for enhanced entry into cells by keeping RBD in a standing position which enhance the virus binding to host cell membrane ACE2. In order to fuse the membrane, the SRAS-CoV spike needs to be proteolytically activated at the S1/S2 boundary to dissociate S1, and S2. The cleavage of the S1–S2 protein, which is required for the conformational changes of the S2 subunit and processing of viral fusion, is regulated by the host proteases, including TMPRSS2 (independently) and cathepsin L (during endocytosis). The TMPRSS2 may cleave the S protein [26]. The viral spike protein mediates SARS-CoV-2 entry into host cells and harbors a S1/S2 cleavage site containing multiple arginine residues (multibasic) not found in closely related animal coronaviruses [27]. After protease cleavage of the S1 protein, the FP subunit of S2 undergoes membrane fusion with the host cell membrane [28,29]. These entrance characteristics of SARS-CoV contribute to its rapid spread and result in severe symptoms and high mortality in infected patients [30].

After entry into the host cell, SRAS-CoV-2 lowers the ACE2 expression, which in turn regulates angiotensin II (Ang II). ACE2 is a type 1 integral membrane glycoprotein that is constitutively expressed by the epithelial cells of the lungs, kidneys, intestine, and blood vessels. In normal physiology, ACE2 breaks down Ang II and, to a lesser extent, angiotensin-I (Ang I) to smaller peptides, angiotensin 1–7 and angiotensin 1–9, respectively [31]. ACE2/Ang 1–7 system plays an important anti-inflammatory and anti-oxidant role protecting the lung against ARDS; indeed, ACE2 has been shown to be protective against lethal avian influenza A H5N1 infection [32].

The ACE2 molecule, besides being a receptor of SARS-CoV and SARS-CoV-2, reduces the activity of the renin–angiotensin system (RAS) by converting Ang I and Ang II into Ang 1–9 and Ang 1–7 respectively [33]. Thus, the ACE2 protein has been shown to play an important role in protecting against some disorders such as cardiovascular complications, chronic obstructive pulmonary disease (COPD) and diabetes, among other COVID-19 comorbidities [34]. The ACE2/Ang 1–7 axis counterbalances the ACE/Ang II-I axis by decreasing Ang II levels, the activation of angiotensin type 1 receptors (AT1Rs) and, thus, leads to decreased pathophysiological effects on tissues, such as inflammation and fibrosis [35].

Ang II interacts with its receptor, Ang II receptor type 1 (AT1R), and modulates the gene expression of several inflammatory cytokines via nuclear factor κB (NF-κB) signaling. This interaction also promotes macrophage activation and results in the production of inflammatory cytokines that may cause ARDS or macrophage activation syndrome (MAS). Some metalloproteases, such as ADAM17, cleave these pro-inflammatory cytokines and ACE2 receptors, resulting in their release as soluble forms. This contributes to the loss of the protective function of surface ACE2 and potentially exacerbates SARS-CoV-2 pathogenesis [36]. SARS-CoV-2 infects the mononuclear phagocyte system, the cells in which produce different pro-inflammatory cytokines to trigger focal inflammation and systemic inflammatory response, a phenomenon known as a cytokine storm. Altogether, these events play a fundamental role in severe presentations of COVID-19, including ARDS and death [37].

#### Role of Pulmonary Alveolar Type II Epithelial Cells and Macrophages in SARS-CoV-2 Infection

As mentioned, SARS-CoV-2 infects alveolar type II epithelial cells and downregulates ACE2 expression, leading to the upregulation of and metabolic dysfunction in Ang II. Ang II acts on the AT1R, activating macrophages and releasing various inflammatory cytokines that trigger tissue inflammation and destruction. As type II alveolar cells, these macrophages also express furin and TMPRSS2, which are responsible for SARS virus exposure [38,39], as well as ADAM17, which acts as a sheddase of ACE2 [40]. Theoretically, after the invasion of macrophages and type II alveolar cells, viruses replicate quickly within macrophages and dendritic cells and trigger the aberrant production of pro-inflammatory cytokines [41] (Figure 1). However, recent studies have detected the increased expression of pro-inflammatory chemokines in human macrophages even in the absence of SARS-CoV-2 replication [42] or antiviral cytokine production [43]. Pulmonary dendritic cells also exert anti-inflammatory effects through antigen presentation and regulation of T cell reactions. Notably, a study revealed that human dendritic cells are susceptible to SARS-CoV-2 infection and cannot maintain viral replication [34].

### 2.2. Innate Immune Responses

#### 2.2.1. Local (Pulmonary) Innate Immune Responses to COVID-19 Infection

After harboring SARS-CoV-2, alveolar epithelial cells initiate innate immunity within the lungs [45]. Viral RNA, which constitutes a pathogen-associated molecular pattern (PAMP), is detected by various sensors including toll-like receptors (TLRs) 3, 7, and 8, as well as retinoic acid-inducible gene I-like receptors [46]. TLRs upregulate antiviral and pro-inflammatory mediators and trigger NF-kB signaling-mediated inflammatory pathways within the lungs [47]. In alveolar epithelial cells and macrophages infected with SARS-CoV-2, NF-κB contributes crucially to the production of various inflammatory cytokines and the development of cytokine storms [48]. In a recent study involving meta-transcriptomic sequencing, a robust innate immune response, hypercytokinemia, and expression of interferon (IFN)-stimulated genes (ISGs) in the bronchoalveolar lavage fluid were found in patients with COVID-19 [49]. Considering that SARS-CoV-2 robustly triggers the expression of numerous ISGs with immunopathogenic potential, with overrepresentation of genes involved in inflammation, ISGs can be used to determine disease severity.

#### 2.2.2. Systemic Innate Immune Responses against COVID-19 Infection

T cell activation and differentiation occur after macrophage processing and the presentation of SARS-CoV-2 particles to T cells. This is followed by immune response amplification, as indicated by a massive release of cytokines [50], including interleukins (ILs) 1, 6, 8, and 21, as well as tumor necrosis factor-β and CCL2. Subsequently, activated lymphocytes and leukocytes are recruited to the site of infection [50]. Viral infection further induces the expression of cathelicidin and defensins, the antimicrobial peptides within the infected cells. Human cathelicidin peptide LL-37, a small, cationic, and amphipathic particle, facilitates the effective binding of cathelicidin and prevents viral invasion [51,52]. During antimicrobial attacks, cathelicidin removes the outer membrane of the virus through a single event rather than a gradual process [53]. This causes the leakage of viral components, which in turn leads to the death of the virus [54,55]. A study on Venezuelan equine encephalitis virus infections reported that cathelicidin inhibited virus entry and modulated the expression of IFN-β1 expression in the infected host cells, eliciting an antiviral response through the inhibition of viral replication [56]. Defensin, another antimicrobial peptide associated with the first line of immunological defense [57], also suppresses viral infection, either through the direct blockage of viral particles or indirectly through the indirect disruption of the viral life cycle [58]. Growing evidence suggests that antiviral activity related to defensins occurs not only during viral entry [59] but also modifies innate immune responses to viral infections. The most important mechanisms of these responses include the activation of T cells, recruitment of macrophages and dendritic cells, differentiation and maturation of dendritic cells, and production of pro-inflammatory cytokines by macrophages and mast cells [60] (Figure 2).

### 2.3. Adaptive Immune Response: T Cell Differentiation and Inflammatory Cytokines in COVID-19

Compared with COVID-19, fewer cases of severe Middle East Respiratory Syndrome coronavirus and SARS-CoV cases result in fatal lower respiratory tract infections and systemic extrapulmonary manifestations [61]. Cytotoxic T cells account for approximately 80% of inflammatory cells related to SARS-CoV in the pulmonary interstitium. Thus, they play a vital role in clearing the viral particles and are responsible for most immune-related injuries [50]. Adaptive immunity occurs through activation of CD4^+^ T cells, which promotes the production of virus-specific antibodies by B cells. Depletion of CD4^+^ T cells results in the reduced production of neutralizing antibodies and cytokines in patients with interstitial pneumonitis [62]. A 2020 study reported that T cell activation was substantially higher in patients with COVID-19 pneumonia, and that the T cells preferentially differentiate into Th17 cells [63].

The persistent response of T cells to S proteins and other structural proteins (M and N proteins) in SARS coronavirus is well established. This serves as a valuable reference for the development of SARS vaccines, particularly with respect to the induction of long-term memory in T cell and immune responses. However, another report suggested that protective antibodies against SARS-CoV-2 infection may not last long [64]. In another study, serological surveys in early convalescence revealed reduced levels of immunoglobulin G and neutralizing antibodies [65]. Clinical studies have indicated that the level of naïve CD4^+^ T cells, regulatory T cells (Tregs), and Th2 cells is higher in patients with poor prognosis than that in patients with good prognosis, as reflected by progression in disease severity [66,67]. Immunosuppression and an enhanced inflammatory response were implicated in disease progression among patients with poor prognosis [68].

On the other hands, dendritic cells exhibit anti-inflammatory activity through antigen presentation and the regulation of T cell reactions to SARS-CoV. This will decrease the degree of CD4^+^ T cells induce pro-inflammatory cytokine production through the activation of the NF-kB pathways [48]. They also produce IL-17, which recruits more inflammatory cells to the infection site, with further activation of downstream cytokine and chemokine cascades [50,69].

### 2.4. Renin–Angiotensin System and COVID-19

The activation of the renin–angiotensin–aldosterone system (RAS) and Ang-II-related inflammation and fibrosis play vital roles in COVID-19 infection and mortality.

Renin converts angiotensinogen into angiotensin I, which is again metabolized to Ang II by the dipeptide carboxypeptidase ACE. The pro-inflammatory effects of Ang II [70] are exerted in concert with the AT1R. In a recent study, the ACE2 receptor and the downstream signaling pathway were identified as an essential counter-regulatory mechanism to RAS activation. Aldosterone reduces membrane ACE2 expression. Under favorable conditions, Ang II can be converted to Ang 1–7 via ACE2, the counter-regulatory effects of which are mediated by the Mas receptor [71].

Strains of both SARS-CoV and SARS-CoV-2 have been shown to use ACE2 receptors and enter the affected cells [72]. The identification of ACE2 and its modulation on the RAS constitutes an interesting topic for the development of therapeutic targets [73]. ACE2 exists in both membrane-bound and soluble forms, and SARS-CoV-2 infection involves the binding of the S protein to the membrane form [74]. A study reported that poor prognosis in SARS-CoV infection was accompanied by ACE-2 downregulation [75]. The virus attaches to ACE-2; the complexes then enter the cells by endocytosis. Viral complexes that are not endocytosed are digested by ADAM17 and lead to critical illness [76]. Less entry of viral particles into cells is associated with better clinical outcomes. A viewpoint paper (2020) revealed that reduced ACE-2 levels are correlated with more severe clinical presentations and harmful end organ damage [77].

## 3. Role of Vitamin D in the COVID-19 Era

### 3.1. Antiviral Activity of Vitamin D and the Innate Immune Response

The promotion of antiviral immunity by vitamin D, which is of great relevance to the current discussion on COVID-19, involves various mechanisms that overlap with antibacterial responses, such as the induction of cathelicidin and defensins, which can block viral entry into cells as well as suppress viral replication [65,78]. Another property of vitamin D relevant both to antibacterial and antiviral mechanisms acts through the promotion of autophagy [79], a fundamental biological process that maintains cellular homeostasis through the encapsulation of damaged organelles and misfolded proteins by intracellular membranes. Autophagy is also an essential mechanism by which cells respond to viral invasion. Specifically, autophagic encapsulation packages viral particles for lysosomal degradation and subsequent antigen presentation and activation of adaptive antiviral immune responses [80]. Thus, autophagy facilitates the creation of a cellular environment that is hostile to viruses but does not guarantee one.

In addition to its established function in bone homeostasis, vitamin D modulates and regulates multiple processes, including host defense, inflammation, immunity, and epithelial repair. Patients with respiratory disease are frequently presented with deficient in vitamin D; supplementation might provide substantial benefits to this population [81]. After binding to serum vitamin D binding protein, circulating 25-hydroxyvitamin D enters monocytes and increases the intracellular level of active 1,25-dihydroxyvitamin D (1,25D), which after binding to vitamin-D receptor (VDR) induces the expression of antimicrobial peptides cathelicidin and β-defensin 4A and promotes autophagy through autophagosome formation [82]. In humans, cathelicidin [83] and β-defensin [84] are produced through a vitamin D-dependent antimicrobial pathway. Our previous study also demonstrated that vitamin D-treated uremic hyperparathyroidism can efficiently increase serum cathelicidin levels [85]. Taken together, vitamin D promotes innate immunity through the expression of both cathelicidin and β-defensin, enhances autophagy through TLR activation, and affects complement activation. Figure 2 presents the putative immune-related mechanisms of vitamin D linked to COVID-19.

### 3.2. Vitamin D Regulates Adaptive Immunity

The adaptive immune system is initiated by the activation of antigen-presenting cells, such as dendritic cells and macrophages, which in turn activate the antigen-recognizing cells, T lymphocytes and B lymphocytes, which are major determinants of the immune response [86]. 1α,25-Dihydroxyvitamin D directly modulates inflammatory cytokines that are dependent on NF-κB activity in numerous types of cells, including macrophages, by blocking NF-κB p65 activation via the upregulation of the NF-κB inhibitory protein IκBα [87]. TLRs are transmembrane proteins that recognize conserved molecular motifs of viral and bacterial origin and initiate innate immune responses. TLR3 recognizes viral double-stranded RNA or synthetic double-stranded RNA (polyinosinic: polycytidylic acid) and is primarily involved in viral defense. Vitamin D treatment has been demonstrated to attenuate the expression of IL-8 in respiratory epithelial cells by double-stranded RNA-TLR3 [86,88].

Circulating T cells, B cells, and dendritic cells express the vitamin D-activating enzyme CYP27B1 (1α-hydroxylase) and the VDR, which then utilize the circulating 25D through intracrine conversion to active 1,25D. Increased intracellular 1,25D inhibits the maturation of dendritic cells and thus modulates the function of CD4^+^ T cells. Systemic active vitamin D (1,25D) also regulates VDR-expressing CD4^+^ T cells in a similar way. In essence, vitamin D inhibits the activation of type 1 T helper cells and cellular immune responses related to tissue destruction. In addition, vitamin D promotes the association of Th2 cells with humorally mediated immunity. In general, vitamin D regulates immunity by promoting the shift from Th1 to Th2 cells. Vitamin D also mitigates inflammation and tissue damage by inhibiting the development of Th17 cells. Similarly, Tregs suppress inflammation in response to vitamin D [82]. In short, vitamin D was assumed to modulate adaptive immunity against COVID-19 in several ways. For example, it can suppress the maturation of dendritic cells and weaken the antigenic presentation, and then increase cytokine production induced by CD4^+^ T cells and promote the efficiency of Treg lymphocytes. Recent clinical study revealed that severe immunosuppression but not a prominent cytokine storm characterizes COVID-19 infections [68]. Vitamin D can also suppress the secretion of Th1 and Th17 cytokines and related tissue destruction (Figure 3), these effects are hypothesized to occur even during COVID-19, suggesting that appropriate vitamin D supplementation may attenuate immunosuppression and enhance anti-inflammatory effects against COVID-19.

### 3.3. Vitamin D Modulates ACE2 and the RAS

Vitamin D deficiency is a known pandemic and global public health problem which varies with age, ethnicity, and latitude. Environmental factors, including the lack or reduction of sunlight (UV-B), life with air pollution, and smoking, are responsible for vitamin D deficiency. The presence of comorbid diseases, such as septicemia, diabetes mellitus, chronic respiratory diseases, and cancer, is closely linked to vitamin D deficiency [89]. In this COVID-19 pandemic, a similarity in prevalent areas and the nature of SARS-CoV-2 infection and vitamin D deficiency was observed [9], which might explain the importance of vitamin D supplementation in COVID-19 [90]. Adequate vitamin D supplementation is also required to reduce RAS activity and increase ACE2 concentrations in acute lung injury. Specifically, sufficient vitamin D supplementation induces the ACE2/Ang 1–7 and suppresses the renin axis and the ACE/Ang II/AT1R axis [91].

It is well known that lower levels of 25 hydroxyvitamin D (25[OH]D) are strongly and independently associated with an increased risk of developing high blood pressure [92]. In line with this finding, both animal and human studies suggest that vitamin D deficiency may increase RAS activity within the kidneys both systemically and locally [93]. A study reported that individuals with vitamin D insufficiency and deficiency (15–29.9 ng and <15 ng/mL, respectively) had higher circulating Ang II concentrations than did those whose 25(OH)D concentrations were sufficient (≥30 ng/mL) [94]. The patients deficient in vitamin D also had a significantly reduced renal plasma flow response to Ang II infusion [95].

The COVID-19 prognosis of older adults, smokers, and individuals with obesity or other comorbidities such as hypertension and diabetes mellitus is poor [96]. RAS-acting agents that increase ACE2 levels serve as substrates for SARS-CoV-2 infection [97]. Circulating ACE2 is regarded as a biomarker of hypertension and heart failure [98] as well as diabetes [99]. SARS-CoV-2 infection downregulates ACE2 activity and accumulates toxic Ang II and metabolites, which subsequently develop into ARDS or fulminant myocarditis [97]. Vitamin D sufficiency can lower RAS activity through several pathways, including transcriptional suppression of renin, ACE, and Ang II expression [100] and increased ACE2 concentration in lipoprotein (LPS)-induced acute lung injury(ALI) [91]. In other words, vitamin D mitigates LPS-induced ALI by inducing the ACE2/Ang 1–7 axis and by suppressing both renin and the ACE/Ang II/AT1R axis [91]. Vitamin D treatment also increases soluble ACE2 (sACE2) [101] which retains the enzyme activity of ACE2 and can bind to the S-protein of SARS-CoV. So sACE2 may block S protein and prevent cells from being infected. In vivo studies on diabetic rats supplemented with active vitamin D reported that calcitriol decreased the ACE concentration and ACE/ACE2 ratio and increased ACE2 concentration [102]. Expression of ACE2 is reduced in patients with DM possibly due to glycosylation [31,103]; this could explain the increased predisposition to severe pulmonary lesions and ARDS with COVID-19. As a result, we can speculate on the beneficial effect of the vitamin-D supplement on diabetic patients with COVID-19.

Vitamin D increases the expression and bioavailability of ACE2, a mechanism that may be responsible for the trapping and inactivation of viruses. This suggests the potential benefits of adequate vitamin D supplementation, which requires further exploration. In sum, vitamin D may be able to combat COVID-19 and the related induction of MAS and ARDS by targeting ACE2 downregulation and the unbalanced RAS (Figure 4).

## 4. Controversial Findings from Clinical Studies

In a recent study on 20 European countries, a close association was found between lower levels of vitamin D and higher numbers of COVID-19 cases and mortality [2]. Vitamin D insufficiency or deficiency is more common in patients with obesity and diabetes, conditions that appear to lead to higher COVID-19 mortality rates [104]. However, recent data from a UK Biobank study did not reveal a relationship between vitamin D levels and the risk of SARS-CoV-2 infection. Moreover, vitamin D insufficiency or deficiency could not explain the ethnic differences in SARS-CoV-2 infection rates [105]. Although it is unclear whether vitamin D can help mitigate the risk of COVID-19 infection or its outcomes, it can be concluded that vitamin D deficiency is not beneficial. A 2020 retrospective cohort study conducted at an urban academic medical center indicated that vitamin D deficiency was associated with increased COVID-19 incidence [5]. Although epidemiological data appear to link vitamin D with COVID-19, conclusive evidence regarding the role of vitamin D in preventing or mitigating the severe respiratory complications of COVID-19 is lacking [106]. Multiple observation studies [107] have shown that serum concentrations of 25 hydroxyvitamin D (25(OH)D) are inversely correlated with the incidence or severity of COVID-19. Evidence to date has generally met Hill’s causality criteria [108] in a biological system, although experimental verification is not sufficient. Vitamin D deficiency constitutes only one of numerous determinants of COVID-19 outcomes, but it can be corrected safely and inexpensively [108,109]. Evidence for the contribution of vitamin D to reducing the risk of COVID-19 includes the speculation that the outbreak is more serious in cooler regions and during the winter months, when 25(OH)D levels are the lowest, as well as the fact that case fatality rates from the disease are higher in patients who are potentially deficient in vitamin D (e.g., older adults and those with comorbidities) [6]. The fact that people tend to gather indoors in cooler regions and winter months versus outdoors in summer may account for the higher COVID-19 incidence observed in winter. Thus, ensuring adequate vitamin D intake (in line with national recommendations) probably be advisable.

Specifically, vitamin D supplements at therapeutic doses might not be dangerous to COVID-19 patients; it might even mitigate the exacerbation or severity of the disease. Of course, considering the emergency of the current situation, sufficient evidence to support this claim has yet to be collected. The recommended intake for individuals at higher risk of COVID-19 is 10,000 IU/day for 1 to 2 weeks and subsequently 5000 IU daily, with target 25(OH)D concentrations of over 40 to 60 ng/mL [6]. Well-conducted randomized controlled trials of vitamin D supplementation in patients with COVID-19 remain an urgent need.

## 5. Conclusions

Vitamin D levels are closely associated with COVID-19 severity and mortality. Adequate vitamin D supplementation may enhance the innate immune response against the disease through the increased synthesis of antimicrobial peptides to kill the virus intracellularly (e.g., intravesicularly). Regarding adaptive immunity, in addition to suppressing cytokine storms, vitamin D enhances CD4^+^ T cells and promotes the production of virus-specific antibodies by activating T cell-dependent B cells. Vitamin D induces ACE2/Ang 1–7/MasR axis activity and inhibits the renin and ACE/Ang II/AT1R axis, which increases the concentration of ACE2, MasR, and Ang 1–7, and subsequently reduces RAS-related tissue inflammation, destruction, and fibrosis. The increased shedding of soluble ACE2 may provide additional protective effects. Clinical evidence on vitamin D supplementation in patients with COVID-19 remains inconclusive, perhaps due to varied study end-points, variations in epidemiological characteristics (e.g., race and dietary habits) and differed clinical settings. However, it is cost effectiveness to give vitamin D to boost immunity from prevention in COVID-19 era. It is important to conduct more randomized studies among different geographical populations to determine variations in requirements of vitamin D supplementation during COVID-19 era.

## Figures and Tables

**Figure 1 ijms-22-05251-f001:**
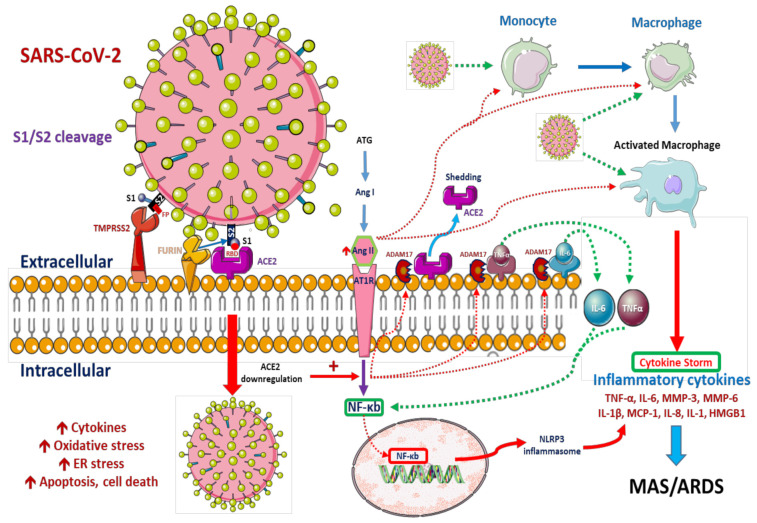
Possible pathophysiological pathways after the entry of severe acute respiratory syndrome coronavirus 2 (SARS-CoV-2). SARS-CoV-2 infects both alveolar macrophages and type II alveolar cells by binding to angiotensin-converting 2 (ACE2) receptors. Before SRAS-CoV-2 enters the host cells, the spike protein 1 (S1) should be pre-activated by the host furin, a convertase proprotein, which will expose the receptor binding domain (RBD) of S1. RBD has a strong binding affinity for the host cell membrane ACE2 for effective entry. After binding of RBD and ACE2, the type 2 transmembrane protease (TMPRSS2) will proteolytically activated the S1/S2 boundary through cleavage the S1–S2 protein which will cause drastic structural changes with further expose the fusion peptide (FP) of S2 which will facilitate the processing of viral-host cell fusion [44]. Immediately after entry, S protein activation is mediated by lysosomal cathepsins and/or furin within the TGN [14,15]. SARS-CoV-2 replication is suppressed by synthetic furin inhibitors [16]. After entry into the host cell, the virus downregulates ACE2 expression, which in turn upregulates angiotensin II (Ang II). Ang II interacts with its receptor, Ang II receptor type 1, and modulates the gene expression of several inflammatory cytokines via nuclear factor κB signaling. This interaction also promotes macrophage activation and results in the production of inflammatory cytokines that may cause acute respiratory distress syndrome or macrophage activation syndrome. Some metalloproteases, such as ADAM metallopeptidase domain 17, cleave these pro-inflammatory cytokines and ACE2 receptors, resulting in their release as soluble forms. This contributes to the loss of the protective function of surface ACE2 and potentially exacerbates SARS-CoV-2 pathogenesis [39]. Monocytes and macrophages in the mononuclear phagocyte system, infected with SARS-CoV-2, produce various pro-inflammatory cytokines and chemokines, a process critical for the induction of local and systemic inflammatory responses known as cytokine storms [18].

**Figure 2 ijms-22-05251-f002:**
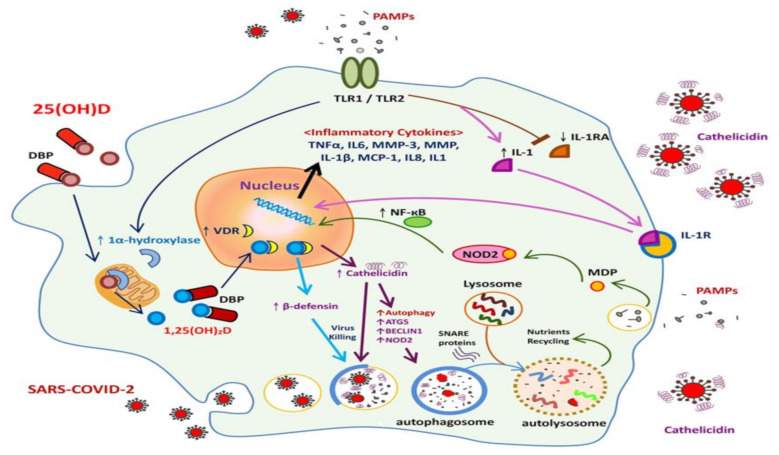
Putative vitamin D-related innate immunity (anti-infection activity) and autophagy responses to coronavirus disease 2019 (COVID-19) infection. The activation of monocyte toll-like receptors (TLR1/TLR2) by pathogen-associated molecular patterns (PAMPs) induces the expression of the cytokine interleukin-1 (IL-1) and suppresses the expression of the IL-1 receptor antagonist, thereby enhancing intracrine signaling by IL-1 and increasing the activity of nuclear factor кB (NF-кB). Pathogen phagocytosis increases the intracellular concentrations of muramyl dipeptide (MDP), which can then bind to the intracellular pathogen recognition receptor NOD2 and increase NF-кB activity. In addition, the activation of TLR1/TLR2 by PAMP results in the transcriptional induction of vitamin D receptor (VDR) and the activation of 1α-hydroxylase expression. Circulating 25-hydroxyvitamin D [25(OH)D] bound to serum vitamin D-binding protein enters monocytes in its free form and is converted to active 1,25-dihydroxyvitamin D [1,25(OH)2D] by mitochondrial 1α-hydroxylase. It then binds to VDR and acts as a transcription factor, induces the expression of cathelicidin and β-defensin 4A, and promotes autophagy through autophagosome formation. NF-кB also enhances the transcriptional induction of cathelicidin and β-defensin 4A. In the presence of increased cathelicidin, immune cells induce the activity of NOD2/CARD15-β-defensin 2, autophagy-related protein 5 (ATG5), and BECLIN1, and they then induce autophagy. Cathelicidin, β-defensin 4A, and mature autophagosomes then work in concert to eliminate bacteria. Cytoplasm SNAP receptor proteins mediate fusion between autophagosomes and lysosomes, and various lysosomal enzymes further hydrolyze proteins, lipids, and nucleic acids. Digestive nutrients may be recycled and utilized by the cells. The net efficacy of such a response is highly dependent on vitamin D status, as well as the availability of circulating 25(OH)D for intracrine conversion to active 1,25(OH)2D by the enzyme 1α-hydroxylase. Activation of TLR1 and TLR2 by PAMP induces the expression of cytokines and inflammatory pathways. Adequate vitamin D supplementation may strengthen the innate immune response against COVID-19 through TLR activation and autophagy, enhance antimicrobial peptide synthesis, and increase the generation of lysosomal degradation enzymes within macrophages.

**Figure 3 ijms-22-05251-f003:**
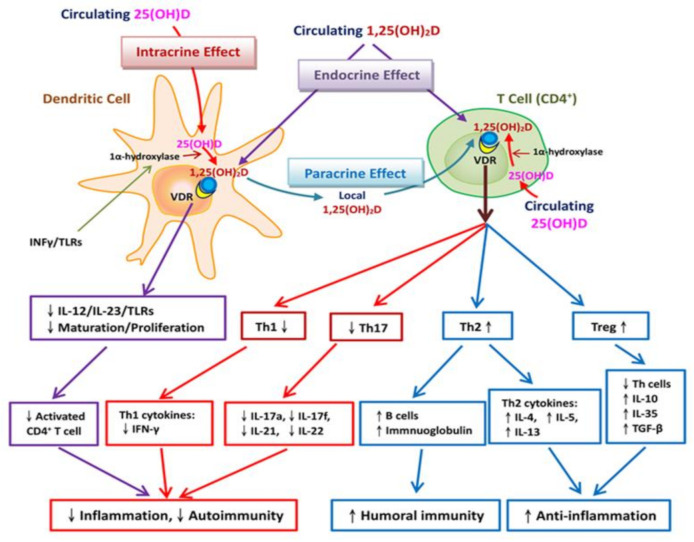
Vitamin D-related adaptive immune responses to COVID-19. Dendritic cells expressing 1α-hydroxylase and the vitamin D receptor (VDR) can utilize circulating 25-hydroxyvitamin D [25(OH)D] for intracrine responses through localized conversion to active vitamin D [1,25(OH)2D]. Intracrine synthesis of 1,25(OH)2D inhibits the maturation of dendritic cells, thereby modulating CD4^+^ T cell function. CD4^+^ T cell responses to 25(OH)D may also be mediated in a paracrine manner, with 1,25(OH)2D acting on VDR-expressing CD4^+^ T cells. VDR-expressing CD4^+^ T cells are also potential targets for systemic 1,25(OH)2D (endocrine effect). Vitamin D acts on dendritic cells to stimulate effector CD4^+^ cells to differentiate into one of the four types of CD4^+^ cells. Activated T cells also express VDR. Under normal circumstances, vitamin D increases T helper (Th) 2 (Th2) cytokines (e.g., IL-10) and the efficiency of regulatory T (Treg) lymphocytes. Vitamin D inhibits the development of Th1 cells, which are associated with the cellular immune response. In addition, vitamin D promotes the association of Th2 cells with humorally mediated immunity. Thus, vitamin D promotes the shift from Th1 to Th2 cells. Vitamin D also inhibits the development of Th17 cells, which play roles in tissue damage and inflammation. The fourth group of CD4^+^ T cells, Tregs, suppress the function of vitamin D. Circulating and local active vitamin D acts through intracrine, paracrine, and endocrine effects to regulate adaptive immunity in SARS-CoV infection. First, it suppresses the maturation of dendritic cells and weakens the antigenic presentation. Second, it increases cytokine production by CD4^+^ T cells and promotes the efficiency of Treg lymphocytes. Finally, it suppresses Th1 and Th17 cytokine secretion, as well as related tissue destruction [82].

**Figure 4 ijms-22-05251-f004:**
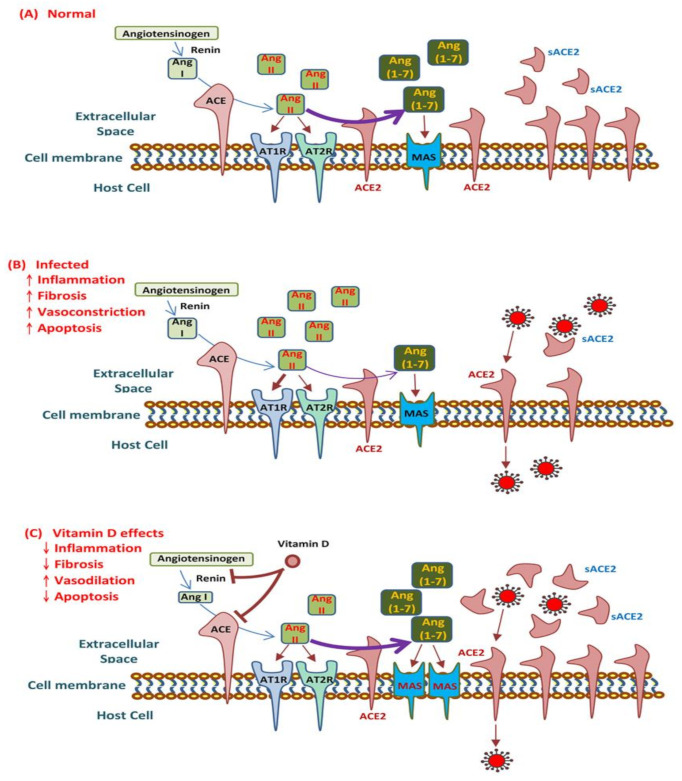
Effects of vitamin D on angiotensin-converting enzyme 2 (ACE2) and the renin–angiotensin–aldosterone system (RAS) in response to the coronavirus disease 2019 (COVID-19). (**A**) Schematic of the RAS under normal circumstances, with physiological steps of the generation of angiotensin (Ang) II and Ang 1–7 shown, as well as their activity on specific receptors. (**B**) Interaction of the severe acute respiratory syndrome coronavirus 2 with the RAS. (**C**) Possible therapeutic effects of vitamin D for COVID-19 and related acute respiratory distress syndrome or macrophage activation syndrome. The ACE2 molecule, besides being a receptor of SARS-CoV-2, reduces the activity of the renin–angiotensin system by converting Ang I and Ang II into Ang 1–9 and Ang 1–7 respectively [33]. Thus, the ACE2 protein has been shown to play an important role in protecting against some disorders such as cardiovascular complications, chronic obstructive pulmonary disease (COPD) and diabetes, among other COVID-19 comorbidities [34]. The ACE2/Ang 1–7 axis counterbalances the ACE/Ang II-I axis by decreasing Ang II levels, the activation of angiotensin type 1 receptors (AT1Rs) and, thus, leads to decreased pathophysiological effects on tissues, such as inflammation and fibrosis [35].

## Data Availability

This is a narrative review article. The primary collection of documents for analysis and review comes from PubMed.

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
