# Peer review of "Immunological Aspects of SARS-CoV-2 Infection and the Putative Beneficial Role of Vitamin-D"

_ijms, 2021, doi:10.3390/ijms22105251_

Round 1
Reviewer 1 Report
The Authors review the several mechanisms through which innate and adaptive immune system interact and react with SarsCov-2 and they describe a “possible” role for vitamin D.
The manuscript is too long and out of focus, particularly in the part describing SarsCov-2 and immune reactions (not related to vitamin D) that represents about 85% of the article. I would suggest to only briefly mention these mechanisms and to focus on vitamin D effect.
Importantly, it seems that the hypotheses are only based on speculations derived from previous studies not inherent COVID19; and that no biological data about specific studies on vitamin D and SarsCov-2 infection are available. If true, this should be clearly stated in the abstract, introduction, and conclusion not to mislead the reader. In fact, paragraph 4 reports some “true” evidence on vitamin D and COVID19 that seem to weaken the various “hypotheses” stated before.
Specific
- Abstract: “In conclusion, vitamin D defends the body against SARS-CoV-2 through a novel complex mechanism that operates through interactions between the activation of both innate and adaptive immun- 55 ity, ACE2 expression, and inhibition of the RAS system. Adequate vitamin D supplementation may be essential for mitigating the progression and severity of COVID-19.” It seems that these conclusions come from data on vitamin D and COVID19 studies. If it is not like this and they are speculations please be clear.
- Introduction is very nice and proper.
- Lines 93 to 349: vitamin D is not mentioned. This large part of the manuscript and inherent figures are very nice, but completely unfocused and should be summarized in a small paragraph with maximal a figure. Otherwise the review title should change into something like “immunonological aspects of SarsCov-2 infection, including the role of vitamin D”
- Line 377 and figure 2: the Authors properly speculate that the above mentioned immune effects mediated by vitamin D may be active even in COVID-19 infection. Correct?
- Line 403 onwards “In short, vitamin D modulates adaptive immunity against COVID-19 in several ways…” Again it seems that these statements are taken from other studies and not from assays on vitamin D effect on Th subpopulations in patients with COVID-19. This should be clear and it should be stated that these effects are hypothezised to occur even during COVID-19.
- Lines 415: “Environmental factors, including the lack or reduc- 415 tion of sunlight (UV-B), life with air pollution and smoking, are attributed to vitamin D 416 deficiency” these factors may “account for” not “be attributed to” vitamin D deficiency.
- Line 423 “In other words” is better stated as “Specifically,..”.
- Line 444 “Vitamin D treatment 444 also increases soluble ACE2 (sACE2) [106] which retains the enzyme activity of ACE2 and 445 can bind to the S-protein of SARS-CoV” another hypothesis stated as a demonstrated thing.
- “In vivo studies on diabetic rats supplemented with active vitamin D 447 referenced in a review article reported that calcitriol decreased the ACE concentration and 448 ACE/ACE2 ratio and increased ACE2 concentration [107]” please cite the original article and discuss why it may be inherent with SarsCoV-2.
- Numbers of paragraphs are not correctly ordered
- Line 483 “it can BE corrected safely and inexpensively”.
Author Response
The Authors review the several mechanisms through which innate and adaptive immune system interact and react with SarsCov-2 and they describe a “possible” role for vitamin D.
The manuscript is too long and out of focus, particularly in the part describing SarsCov-2 and immune reactions (not related to vitamin D) that represents about 85% of the article. I would suggest to only briefly mention these mechanisms and to focus on vitamin D effect.
Importantly, it seems that the hypotheses are only based on speculations derived from previous studies not inherent COVID19; and that no biological data about specific studies on vitamin D and SarsCov-2 infection are available. If true, this should be clearly stated in the abstract, introduction, and conclusion not to mislead the reader. In fact, paragraph 4 reports some “true” evidence on vitamin D and COVID19 that seem to weaken the various “hypotheses” stated before.
Response:
We are deeply honored by the time and effort you have devoted to the examination of this manuscript. We all learn a great deal from your critiques. The manuscript has been thoroughly edited based on your suggestions.
To allow readers to better understand the various immune mechanisms caused by SARS Cov2 infection, more descriptions of the various immune mechanisms are available. Although there is currently a lack of research findings on randomized double-blind control trials. We try to provide research through clinical observation as well. In order to avoid misleading the readers, we also agree with the reviewer's opinion, we also slightly adjusted and revised the topic into Immunological aspects of SARS CoV-2 infection and the putative beneficial role of vitamin-D. We provided a clearer description of the role of Vit-D among COVID-19 patients. We also add more accurate references in our revised manuscript as suggested by reviewers.
Specific
- Abstract: “In conclusion, vitamin D defends the body against SARS-CoV-2 through a novel complex mechanism that operates through interactions between the activation of both innate and adaptive immun- 55 ity, ACE2 expression, and inhibition of the RAS system. Adequate vitamin D supplementation may be essential for mitigating the progression and severity of COVID-19.” It seems that these conclusions come from data on vitamin D and COVID19 studies. If it is not like this and they are speculations please be clear.
Response: Thank you very much for reviewer’s valuable advice. We had made the necessary rectification according to the reviewer’s suggestions. (Line 56-59, 489-492)
Multiple observation studies [110] have shown that serum concentrations of 25 hydroxyvitamin D are inversely correlated with the incidence or severity of COVID-19. The evidence gathered thus to far generally meet Hill's causality criteria [111] in a biological system, although experimental verification is not sufficient. We speculated that adequate vitamin D supplementation may be essential in mitigating the progression and severity of COVID-19. Future studies are warranted to determine the dosage and efficacy of vitamin D supplementation among different populations of people affected by COVID-19.
- Introduction is very nice and proper.
Response: We are deeply honored by the time and effort you spent in reviewing this manuscript.
- Lines 93 to 349: vitamin D is not mentioned. This large part of the manuscript and inherent figures are very nice, but completely unfocused and should be summarized in a small paragraph with maximal a figure. Otherwise the review title should change into something like “immunonological aspects of SarsCov-2 infection, including the role of vitamin D”
Response: Thank you so much for the valuable feedback from the reviewer. We made the necessary adjustments in accordance with the reviewer’s advice. (Line 2)
To allow readers to better understand the various immune mechanisms caused by SARS Cov2 infection, more descriptions of the various immune mechanisms are available. Although there is currently a lack of research findings on randomized double-blind control trials. We try to provide research through clinical observation as well. In order to avoid misleading the readers, we also agree with the reviewer's opinion, we also slightly adjusted and revised the topic into “Immunological aspects of SARS CoV-2 infection and the putative beneficial role of vitamin-D”.
- Line 377 and figure 2: the Authors properly speculate that the above mentioned immune effects mediated by vitamin D may be active even in COVID-19 infection. Correct?
Response: Thank you so much for your invaluable input. We have revised the statement as reviewer's comments. (Line 262 & 380)
- Line 403 onwards “In short, vitamin D modulates adaptive immunity against COVID-19 in several ways…” Again it seems that these statements are taken from other studies and not from assays on vitamin D effect on Th subpopulations in patients with COVID-19. This should be clear and it should be stated that these effects are hypothezised to occur even during COVID-19.
Response: We are deeply honored by the time and effort you spent in reviewing this manuscript. We have revised the manuscript thoroughly according to your suggestions. (Lin 406, 412-413)
- Lines 415: “Environmental factors, including the lack or reduction of sunlight (UV-B), life with air pollution and smoking, are attributed to vitamin D 416 deficiency” these factors may “account for” not “be attributed to” vitamin D deficiency.
Response: Thank you very much for reviewer’s kind reminder. We have made the revision according to the reviewer’s suggestion. (Line 420)
- Line 423 “In other words” is better stated as “Specifically,..”.
Response: We are deeply honored by the time and effort you spent in reviewing this manuscript. We have revised the statement according to your suggestions. (Line 427)
- Line 444 “Vitamin D treatment 444 also increases soluble ACE2 (sACE2) [106] which retains the enzyme activity of ACE2 and 445 can bind to the S-protein of SARS-CoV” another hypothesis stated as a demonstrated thing.
Response: Thank you very much for reviewer’s valuable comments. The declaration was made to clarify the relationship between the protective effect of vit-D and soluble ACE2. (Line 448-451)
9.“In vivo studies on diabetic rats supplemented with active vitamin D 447 referenced in a review article reported that calcitriol decreased the ACE concentration and 448 ACE/ACE2 ratio and increased ACE2 concentration [107]” please cite the original article and discuss why it may be inherent with SarsCoV-2.
Response: Thanks a lot for the nice reminder from the reviewer. We had revised it based on the views of the reviewer. (Line 451-453)
Expression of ACE2 is reduced in patients with DM possibly due to glycosylation [33, 106]; this could explain the increased predisposition to severe pulmonary lesions and ARDS with COVID-19. As a result, we can speculate on the beneficial effect of the vit-D supplement on diabetic patients with COVID-19.
- Numbers of paragraphs are not correctly ordered
Response: Thank you very much for reviewer’s kind reminder. We have corrected this typo according to the reviewer’s suggestion.
- Line 483 “it can BE corrected safely and inexpensively”.
Response: Thank you very much for reviewer’s valuable advice. We had made the necessary rectification according to the reviewer’s suggestions. (Line 493)

Reviewer 2 Report
I found it very interesting and clear how this work explained the potential crucial role of vitamin D, still discussed, in SARS-CoV-2 virus infection.
Author Response
I found it very interesting and clear how this work explained the potential crucial role of vitamin D, still discussed, in SARS-CoV-2 virus infection.
Response:
We are deeply honored by the time and effort you have devoted to the examination of this manuscript. The manuscript has been thoroughly edited based on your suggestions.
